# The Mechanisms of Zinc Action as a Potent Anti-Viral Agent: The Clinical Therapeutic Implication in COVID-19

**DOI:** 10.3390/antiox11101862

**Published:** 2022-09-21

**Authors:** Ananda S. Prasad, Agnes Malysa, Gerold Bepler, Andrew Fribley, Bin Bao

**Affiliations:** Department of Oncology, Karmanos Cancer Institute, Wayne State University, 4100 John R Street, Detroit, MI 48201, USA

**Keywords:** COVID-19, SARS-CoV2, inflammation, immunity, cytokines, zinc, anti-viral activity

## Abstract

The pandemic of COVID-19 was caused by a novel coronavirus termed as SARS-CoV2 and is still ongoing with high morbidity and mortality rates in the whole world. The pathogenesis of COVID-19 is highly linked with over-active immune and inflammatory responses, leading to activated cytokine storm, which contribute to ARDS with worsen outcome. Currently, there is no effective therapeutic drug for the treatment of COVID-19. Zinc is known to act as an immune modulator, which plays an important role in immune defense system. Recently, zinc has been widely considered as an anti-inflammatory and anti-oxidant agent. Accumulating numbers of studies have revealed that zinc plays an important role in antiviral immunity in several viral infections. Several early clinical trials clearly indicate that zinc treatment remarkably decreased the severity of the upper respiratory infection of rhinovirus in humans. Currently, zinc has been used for the therapeutic intervention of COVID-19 in many different clinical trials. Several clinical studies reveal that zinc treatment using a combination of HCQ and zinc pronouncedly reduced symptom score and the rates of hospital admission and mortality in COVID-19 patients. These data support that zinc might act as an anti-viral agent in the addition to its anti-inflammatory and anti-oxidant properties for the adjuvant therapeutic intervention of COVID-19.

## 1. Introduction

The ongoing pandemic of coronavirus disease 2019 (COVID-19) was caused by a novel coronavirus, officially defined as severe acute respiratory syndrome (SARS) coronavirus 2 (SARS-CoV2) by the World Health Organism (WHO) in February 2020 [1]. In December 2019, the first cases of SARS-CoV2 infection were reported in Wuhan City, China [2,3,4] and were rapidly spreading through the whole world by asymptomatic COVID-19 patients who traveled [1]. By February 21, 2022, more than 423 million cases of COVID-19 have been diagnosed, and more than 5.8 million of COVID-19-related deaths have been documented worldwide by WHO (URL Link: https://covid19.who.int; 21 February 2022). This pandemic is still ongoing without good control of the virus, which is due to (1) the epidemiological features of SARS-CoV2 with a very long incubation time of up to 14 days or more, (2) extended survival time of virus on inanimate material surfaces, (3) viral spread by asymptomatic patients, (4) person-to-person spread via respiratory droplets from or closely contact with COVID-19 patients [5,6,7,8,9,10], and (5) no effective or unproven therapeutic drugs available. Several kinds of vaccines have been reported to reduce on morbidity and mortality. Currently, new cases of COVID-19 patients are still increasing every day and have already caused tremendous burdens to health, social, and economic systems [11].

This pandemic is an unprecedented disaster unlike any we have ever faced in the past several decades, and it presents a significant challenge for us to explore effective interventional strategies for the treatment/prevention of SARS-CoV2 infection. Although approximately more than 20 different therapeutic drugs have been investigated for use in the treatment of COVID-19 in current clinical practices [1,12,13,14], there is no proven, effective therapeutic treatment available. However, all these therapeutic treatments are still under assessment. 

## 2. SARS-CoV2 Virus and Its Pathogenesis

SARS-CoV2 is a novel β-coronavirus, which was discovered in COVID-19 patients in Wuhan, China, in December 2019 [4]. The SARS-CoV2 virus belongs to the sub-family Coronavirrinase in the family of Coronaviridase [15,16] and it is a large enveloped virus with non-segmented, single-strand, and positive-sense RNA genome [15,16]. It has been found that SARS-CoV2 virus has approximately 30,000 nucleotides, encoding about 29 proteins, namely 4 structural proteins (spike glycoprotein, membrane, envelope, and nucleocapsid), 16 non-structural proteins including RNA-dependent RNA polymerase, 3 chymotrypsin-like protease, hemagglutinin-esterase essential for viral replication life cycle, and 9 accessory proteins [12,15,16]. Similar to other novel coronaviruses, such as SARS-CoV discovered in China, 2003 and Middle East respiratory syndrome coronavirus (MERS-CoV) discovered in 2012 [17,18], SARS-CoV2 acts as a primary respiratory pathogen and causes acute respiratory distress syndrome (ARDS). It has been reported that SARS-CoV2 has 70–80% similarity of RNA sequence homology to SARS-CoV and 50–70% similarity to MERS-CoV [13,19,20,21]. All these novel coronaviruses are generally transmitted in humans and animals, and result in ARDS with a high mortality rate. 

Similar to SARS-CoV, MERS-CoV, and other coronaviruses, SARS-CoV2 enters host cells through receptor-mediated endocytosis [22,23,24,25,26,27]. Once it releases from the endosome to the cytoplasm, the virus replicates rapidly in the cytoplasm as virions, which are distributed from contaminated cells to other cells by exocytosis, leading to the death of infected cells [22,23,24,25,26,27]. Currently, it has been discovered that SARS-CoV2 virus primarily enters into host cells with direct binding of the viral structural trans-membrane spike (S) glycoprotein to the peptide domain of angiotensin-converting enzyme 2 (ACE2) in cells such as airway epithelial cells [2,3,23]. Human ACE2 protein has been considered as the cellular receptor for the SARS-CoV and SARS-CoV2 viruses [1,27]. Moreover, the host cluster of differentiation 26 (CD26) has been also considered as a potential receptor for SARS-CoV2 virus, as discussed recently [28]. 

It has been noted that these novel coronaviruses may inhibit the early immune defense system with anti-viral innate cytokines in the body through a viral immune evasion mechanism by the downregulation of MHC molecule expression and inhibition of MHC molecule recognition. This would induce unlimited virus replication cycle and infection amplification, which entails an over-active immune response and the delayed and substantially sustained activation of inflammatory cytokine/chemokine responses at the later stage, eventually causing hyper-inflammation, cytokine storm, and viral sepsis in severe or critical conditions of the patients with novel coronavirus infections [13,29,30,31]. Early and sufficient control of viral replication and pathogen elimination by an innate and adaptive immune system in the body would be a potential target of therapeutic intervention for the prevention/treatment of COVID-19 infection. Zinc is a known immune modulator and can act as an anti-inflammatory and anti-oxidant agent in humans. Several clinical trials indicate that zinc treatment reduces the symptoms and duration of acute respiratory infections in humans. These data suggest that zinc may have a great implication in COVID-19. We will discuss more details of zinc and COVID-19 later in Section 5, Section 6, Section 7 and Section 8. 

## 3. Clinical Manifestation, Activated Cytokine Storm, and Treatment

COVID-19 is typically transmitted by air droplets expelled from patients including asymptomatic patients via close person-to-person contact. Typically, the incubation time of this disease typically ranges from 2–14 days, with an average of 5 days even though some cases have longer than 14 days of incubation [1,32,33]. COVID-19 has a wide range of clinical manifestations from asymptomatic, mild, moderate, severe, or critically ill such as acute lung damage, ARDS, viral sepsis, and even death resulting from respiratory failure or multi-organ failures including cardiovascular, renal, and liver failures. The severity of this disease is associated with high risk for patients over the age of 60 and those with a history of obesity, diabetes mellitus, hypertension, cancers, immune compromised conditions [13,19,28,34,35]. 

Clinical manifestations of COVID-19 patients have been reported in many different clinical studies, as summarized in Table 1 [12,36,37,38]. The degree of medical conditions from mild to severe or critical illness in COVID-19 patients has been reportedly linked with the excessive systemic immune and inflammatory response that entails an activated cytokine storm with a significant increase in inflammatory cytokines/chemokines and molecules, that eventually lead to acute lung damage, ARDS, respiratory failure, multiple organ dysfunctions, or even death [39]. It has been known that this hyper-inflammatory syndrome causes life-threatening ARDS in COVID-19 patients. ARDS patients have a hyper-inflammatory response that is characteristic of lymphopenia, elevated levels of C-reactive protein (CRP), inerleukine-6 (IL-6), fibrinogen, ferritin, and other inflammatory cytokine/chemokines and molecules [39]. 

It has been noted that the leading mortality cause of COVID-19 is acute respiratory failure resulting from ARDS, which is highly linked with the activated cytokine storm due to aberrant inflammatory response by a positive feedback loop induction pattern in the body [1,40]. The activated cytokine storm has been most commonly found in patients due to the hyper-inflammatory response to the acute infection of certain types of microbial pathogens including SARS-CoV, MERS-CoV, and SARS-CoV2 viruses [1,40]. Excessive immune and inflammatory responses are widely believed to be the main driver of pathogenesis of ARDS in SARS-CoV, MERS-CoV, and SARS-CoV2 infections [41,42,43]. 

The characteristics of an activated cytokine storm include abnormal levels of pro-inflammatory cytokines/chemokines and molecules such as IL-1, IL-2, IL-6, IL-7, IL-8, IL-10, IL-12, MCP-1, macrophage inflammatory protein-1 α (MIP-1A), HGF, granulocyte- macrophage colony stimulating factor (GM-CSF), interferon inducible protein-10 (IP-10), CRP, and TNF-α [40,43]. Specifically, evidence shows that ARDS-related deaths have been linked with a significant increase in levels of IL-6 and CRP [40]. Moreover, recent observation studies demonstrate that COVID-19 patients with severe illness had significantly increased levels of inflammatory cytokines/chemokines such as IL-1, IL-2, IL-6, IL-8, IL-10, MCP-1, and TNF-α [1,40,43,44]. Among all the cytokine/chemokines and molecules, IL-1, IL-6, CRP, and TNF-α are commonly considered key inflammatory markers in the acute phase response of infections. 

Importantly, suppressed cellular immune functions such as hypoalbuminemia, lymphopenia, neutropenia, spleen atrophy, and lymph node necrosis have been observed in COVID-19 patients. Moreover, sustained and substantial reduction of subsets of T lymphocytes such as CD4+ helper and CD8+ cytotoxic T lymphocytes have also been found in COVID-19 patients [43]. It has been noted that the reduction of CD4+ and CD8+ T cells is related to pro-inflammatory cytokine/chemokine production [21,42,45]. A significant reduction of CD8+ cytotoxic T lymphocytes, one subset of the key viral killer cells, suggests that COVID-19 patients have dysfunction in viral clearance in the body. Therefore, aberrant immune response, excessive inflammatory response/activated cytokine storm, and suppressed cellular immune function contribute to the pathogenesis of COVID-19 [1,43,44]. However, the pathogenesis of COVID-19 is not fully understood yet.

As mentioned earlier, the severity of COVID-19 is highly linked with super-production of inflammatory cytokines and chemokines, resulting from exacerbation of the cytokine/chemokine cascade in a positive feedback pattern. The aberrant immune response and hyper-inflammation leads to the activated cytokine storm as well as the suppression of cellular immune functions, which leads to uncontrolled viral replication via a viral evasion mechanism in the body, and eventually contributes to severe lung and other tissue damage [1,14,46]. Inhibition of the virus and inflammation as well as the recovery of normal immune function seems more important. Unfortunately, there is no adequate, target-specific therapeutic drugs, which have been approved to be effective for the treatment of COVID 19. Most patients recover within 1–2 weeks without any treatment, but patients with mild symptoms are provided with supportive care or adjunctive treatment. For patients with moderate, severe, or critical illness such as ARDS, a wide variety of repurposed and experimental therapeutic treatments of antiviral drugs such as remdesivir, favipiravir, ribavirin, lopinavir, ritonavir; hydroxychloroquine, and azithromycin; anti-inflammatory drugs such as corticosteroids and cytokine inhibitors; ACE2/angiotensin receptor inhibitors, or convalescent plasma therapy in addition to supportive care and symptomatic therapy are recommended in current clinical practices [1,12,13,14,46]. Other adjuvant treatment including antioxidants and vitamins are also recommended for these patients [1,12,13,14,47]. It has been estimated that more than 15 different anti-viral drugs, and more than 10 different anti-inflammatory drugs have been used in clinical trials, as reviewed recently [1,12,13,14,47]. However, the efficacy of all the currently used drugs for COVID-19 is still under investigation.

## 4. Zinc and Immunity

Zinc is an essential trace mineral element in the body and widely considered as an immune modulator. Evidence shows that zinc plays a wide range of biochemical functions by the involvement of many different biological processes including cell proliferation and differentiation, cell cycle, reactive oxygen species (ROS) homeostasis, and immune response [48,49]. Zinc has been identified to be required for the stabilization of 3D structures of more than 2000 different zinc finger proteins including transcription factors, steroid receptors, hormones, and enzymes in humans [48,49]. Cysteine 2-histidine 2 (C2H2) rich zinc finger domains are involved in regulating expressions of a wide variety of genes encoding growth factors, receptors, immune response mediators, pro-inflammatory cytokines/chemokines, and other proteins by binding DNA, RNAs, and proteins [48,49]. It is noted that viral zinc homeostasis has an important role in viral replications, depending on the context of the viruses. The passive release of bound zinc from the viral zinc finger or zinc containing proteins, termed zinc ejection, has been reported to be associated with the modulation of virial activities. Zinc ejection is caused by the binding of some compounds to viral zinc finger proteins due to redox reactions, resulting in destabilization of these zinc finger proteins, eventually contributing to the inhibition of viral activity. For example, in vitro studies show that zinc ejectors such as disulfide benzamide compounds can bind to HIV nucleocapsid protein (NCp7, a zinc finger protein essential for viral replication), and cause zinc ejection, resulting in the structural destabilizations of this protein, eventually contributing to the inhibition of HIV viral activity [50,51]. Similar results of zinc ejection have been found to be associated with other benzamide compounds by targeting other zinc finger protein in HPV and HCV viruses [52,53]. The addition of zinc may reverse these effects [54]. The phenomenon of zinc ejection has not been reported in novel coronaviruses, which may have different kinds of viral zinc homeostasis. Evidence shows that zinc ionophore compounds such as HCQ can increase the binding of zinc to viral enzymes such as RdRp and 3CL, which contain zinc binding sites, resulting in the inhibition of viral activity in novel coronaviruses. The details of how zinc modulates RdRp and 3CL in novel coronaviruses including COVID-19 virus will be discussed in Section 8.1 and Section 8.2. 

In the 1960′s, Prasad and his team were the first to report zinc deficiency syndrome in the Middle East. The clinical manifestations of this syndrome included growth retardation, anemia, hypogonadism in male patients, and immune response dysfunctions such as increased infections [55]. It was found that this syndrome was caused by diets rich in organic phosphate compounds called as phytates, which impairs the bioavailability of zinc in the body. Iron (Fe) treatment only corrected anemia, while zinc treatment corrected all these symptoms [55]. These early findings strongly suggest that zinc had potential as an essential nutrient in the human body. Subsequently, a large number of experimental and human studies have provided solid evidence to support the essentiality of zinc as a trace mineral nutrient in the human body [56,57,58]. 

Zinc is generally recognized as an essential immune mediator to maintain normal immune defense mechanism against microbial infection in the body. Zinc deficiency has been found to increase infections of a variety of micro-organisms such as viruses, bacterial fungus, and parasites [48,58,59,60]. A large number of experimental and human studies have provided clear evidence to support that zinc plays an essential role through modulation of cellular immune responses including natural killer cells, T lymphocytes, and B lymphocytes [48,57,58,59,60]. For example, zinc can increase NK cell activity, T cell activation, mitogenesis, and antibody synthesis. Our early experimental studies revealed that dietary restriction-induced zinc depletion reduced the relative level of CD8+/CD73+ T lymphocytes in volunteer subjects [61,62]. This subset of T cells is considered as the main precursor of cytolytic T lymphocytes. CD73+ cell surface protein is required in antigen recognition on these cells [57,62]. Furthermore, zinc deficiency causes thymic atrophy along with the down-regulation of thymulin, a zinc sensitive thymic hormone for the regulation of T cell proliferation and differentiation as well as NK cell activation [57,63]. Early experimental studies also revealed that zinc deficient condition (1 μM zinc) suppressed the expressions of IL-2, IL-2 receptor (IL-2R), and IFN-γ in HUT-78 cells (a malignant lymphoid leukemia cell), as compared to zinc sufficient condition (15 μM zinc) [64,65,66,67]. All these findings suggest zinc’s important role in cellular immunity. However, it has been noted that high levels or toxic levels of zinc impair immunity, which is similar to the effect of zinc deficiency on immunity. For example, zinc deficient (1 μM) and zinc toxic (100 μM) conditions decreased the expressions of IL-2, IL-2R-α, and TNF-α in HUT-78 cells, as compared to zinc sufficient or physiological conditions (15 μM). Other documents also support this scenario, in which toxic level of zinc impairs normal immunity [68]. Therefore, zinc homeostasis seems be critical for the maintenance of normal immunity in the body. 

Additionally, emerging numbers of studies have indicated that zinc reduces inflammatory cytokines/chemokines such as TNF-α, IL-1β, IL-6, MCP-1, vascular cell adhesion molecule (VCAM), and CRP as well as oxidative stress markers such as lipid peroxidation products (MDA + HAE) and DNA intermittent oxidation product (8-OHdG) [64,69]. Zinc supplementation reduces the production of inflammatory cytokines/chemokines and oxidative stress markers along with the reduction of infection incidence in children, elderly subjects, and sickle cell anemia (SCD) patients [69,70,71,72,73], which clearly suggest that zinc may act as a potent anti-oxidant and anti-inflammatory agent, potentially through the regulation of multiple molecular and cellular mechanisms including A20, NF-κB, MT, zinc transporters, TTP, and PPAR, as reviewed recently [56,74]. 

As discussed earlier, COVID-19 pathogenesis is closely linked with the dysregulation of immune-inflammatory response in the body. Any candidate drug or agent that relieves patient symptoms using anti-viral drugs may prolong the survival time of patients and may provide a chance for the recovery of the immune defense system leading to the eradication of the virus in patients. Since zinc is non-toxic, non-expansive, readily available, and plays an important role as an immune modulator, its application in COVID-19 may provide a clinical, cost-effective benefit to facilitate a meaningful recovery with a better prognosis. 

## 5. Zinc Deficiency in Acute Respiratory Infection

As mentioned earlier, the alteration of zinc homeostasis or zinc deficiency in the body is highly linked with infections of micro-organisms such as bacteria, funguses, and viruses including respiratory virus infections [49,56]. Zinc deficiency increases infection incidence while zinc supplementation corrects this adverse effect [49,56]. For example, a new study reports that zinc deficiency increased incidence rate of acute upper and lower respiratory infection and mean duration of disease in children, compared to non-deficient groups. Zinc supplementation (20 mg zinc daily; 2 weeks) to these zinc deficient children significantly reversed these adverse effects during a 6-month follow-up period [75], which suggests that zinc may play a protective role in acute virus infection. 

## 6. Zinc Homeostasis in COVID-19 Infection

It is interesting that zinc status in the body is also linked with the severity of COVID-19 infection. It has been reported that pregnant female patients with COVID-19 infection have a lower level of serum zinc during pregnancy, when compared to the non-infected pregnant control women. Zinc level is negatively linked with acute phase markers including IL-6 [76]. Another prospective study reveals that COVID-19 patients have a significant lower level of zinc when compared to the non-infected control group (serum zinc: 74.5 μg/dL vs. 105.8 μg/dL) [77]. A level of 89 μg/dL of serum zinc or less has been considered as zinc deficiency while 90 μg/dL or more considered as zinc sufficiency or normal range in the body [71,78]. Moreover, zinc deficiency has been found to be correlated to the severity of COVID-19 infection such as higher rates of complications, ARDS, steroid therapy, prolong stay of hospitalization, and death [77]. These findings may explain why elderly populations, who are susceptible to being zinc deficient, have a significantly higher risk of COVID-19 morbidity and mortality, as reviewed recently [79]. However, the detailed role of zinc homeostasis in the pathogenesis of COVID-19 is not fully elucidated.

## 7. Zinc Treatment/Supplementation for Virus Infections in COVID-19

It has been widely known that the application of zinc has displayed a clinical benefit in human health and diseases, including in children and elderly subject, and for SCD, cancers, and diabetes [70,80,81,82,83,84]. Accumulating numbers of experimental and clinical studies have provided clear evidence that zinc plays an important role in anti-viral immunity by inhibiting viral replication cycle, RNA synthesis, and protein processes in several viral infections including human immunodeficiency virus-1 (HIV-1), herpes simples virus, human papilloma virus, hepatitis C virus, coronavirus, and varicella-zoster virus, which has been reviewed recently [85], suggesting that zinc may be used for COVID-19 treatment. For example, our early clinical trial study revealed that zinc treatment (13 mg zinc as gluconate zinc for every 2–3 h daily) significantly reduced the overall duration of rhinovirus upper respiratory infection (common cold) (4.0 vs. 7.1 days) in human adults, compared to the placebo-controlled patients. The severity of the disease was significantly decreased in the zinc treatment group [86,87]. Similar findings were observed by other groups [85,88,89,90]. We also observed the mean change in plasma levels of soluble IL-1 receptor agonist (sIL-1ra) in the zinc treatment group whereas there was no change found in the placebo-controlled group. The mean change in plasma level of sICAM showed a significant reduction in the zinc treatment group [87]. One recent meta-analysis including three randomized placebo-controlled trial studies with 199 common cold patients confirmed that the high dose of zinc treatment displays a therapeutic benefit for the treatment of rhinovirus infection in humans [91]. These findings suggest that zinc treatment reduces the severity of rhinovirus infection along with the up-regulation of sIL-1ra, an endogenous IL-1 receptor inhibitor and inhibition of sICAM, an inflammatory cytokine acting as a viral entry receptor. 

Moreover, one clinical trial demonstrated that the combined treatment of HCQ, azithromycin, and zinc decreased the rates of hospital admission (2.8% vs. 15.4%) and death (0.7% vs. 3.4%) of COVID-19 patients, in comparison to untreated control patients [92,93]. It has been reported that the administration of HCQ and azithromycin did not show a rapid anti-viral clearance and any clinical benefit in COVID-19 patients with severe illness [94] even though both drugs have been considered as anti-viral agents for COVID-19 treatment. It is interesting that CQ has been found to act as a zinc-specific ionophore, which enhances cellular zinc bio-availability in vitro. Meanwhile, zinc can increase CQ-induced cytotoxicity, leading to apoptosis of cells in vitro [95]. Therefore, these findings from such triple administrations with zinc suggest that zinc might provide an important role in anti-viral immunity for COVID-19 treatment. According to personal experience, 10 days of treatment of 200 mg HCQ plus 50 mg zinc twice a day can cure symptomatic COVID-19 patients who are over the age of 90, which suggests an important role of zinc in COVID-19 patients who receive HCQ therapy. Moreover, several more clinical trial studies have revealed that zinc or the combination of HCQ with zinc have positive clinical outcomes in the treatment of COVID-19, as summarized in Table 2 [92,93,96,97,98,99,100,101,102],. These data support the positive role of zinc in COVID-19 treatment. The exact mechanism of synergic effect of zinc with HCQ/CQ for COVID-19 treatment is still not fully understood. Currently, more than 18 clinical trials with different doses (ranging from 6 mg to 440 mg) of zinc and treatment periods of time (from 5 days to 2 months) have been documented for the treatment/prevention of COVID-19 as recently reviewed [103,104]. The effectiveness of zinc for COVID-19 is still under investigation. 

The beneficial role of zinc in the prophylaxis or prevention of COVID-19 infection has also received a lot of attention. A case-control study in primary care centers was conducted to examine the effectiveness of zinc supplementation on COVID-19 prophylaxis and treatment. A total of 104 adult volunteers received 15–50 mg zinc for at least 4 weeks as the experimental group while 96 adult volunteers received placebo as the control. The results indicate that zinc supplemented group had significantly lower incidence of symptomatic COVID-19 infection, compared to the placebo group [103]. One prospective, randomized, double-blind study reports that supplementation of doxycycline with zinc (15 mg zinc, 6 weeks) significantly reduces the incidence of COVID-19 infection in health care workers [104]. Another randomized study (open-label) reports that 42 days of supplementation of zinc (80 mg) and vitamin C (500 mg) reduces the incidence of COVID-19 in young healthy subjects (mean age 33 years), compared to control subjects who only received vitamin C (47.3% vs. 70%) [105]. These findings strongly suggest that zinc may play a beneficial role in prophylaxis. However, a well-designed study is required to investigate the detailed role of zinc in prophylaxis of COVID-19 infection. 

## 8. Mechanisms of Zinc as an Antiviral Agent in COVID-19

As mentioned above, the evidence supports that zinc adjuvant therapeutic treatment may have an inhibitory effect on viral infections including SARS-CoV2. However, the mechanism of zinc action as an immune mediator against viral infections is not fully elucidated, even though the application of zinc as an immune modulator for COVID-19 treatment by targeting multiple signaling pathways has been reviewed recently [79,106,107,108]. Here, we will discuss the potential mechanism of zinc as an anti-viral agent for COVID-19 treatment in the following paragraphs.

### 8.1. Viral RNA-Dependent RNA Polymerase (RdRp)

It has been known that viral RdRp is an internal catalytic protein for RNA synthesis from a viral RNA template, which is different from all the organisms because they use DNA as a template for the transcription of RNA. This enzyme is a very critical, non-structural protein (NSP) for the involvement of viral replication life cycle and amplification of infection without the DNA synthesis stage. Such RdRp dependent replication is quick and easy for RNA viruses compared to DNA viruses [12,15,109]. Targeting RdRp would provide a potential therapeutic strategy for the treatment of COVID-19. Several RdRp inhibitors including remdesivir, favipiravir, and ribavirin have been recommended for COVID-19 treatment in current clinical practice. For example, remdesivir has shown a clinical benefit for COVID-19 [110]. 

The evidence has indicated that zinc may inhibit the viral replication cycle of coronavirus in COVID-19 by targeting RdRp. One early in vitro experimental study demonstrated that zinc suppressed RdRp activity in SARS-CoV and Arterivirus (coronavirus) in a dose-responsive pattern (a range from 50–500 μM of zinc), along with the inhibition of viral RNA synthesis and amplification [111]. According to our previous reports, 1 μM of zinc in culture condition was considered as a zinc deficient state. However, 15 μM of zinc or more was considered as physiological or sufficient state of zinc [64,66,67]. This information suggest that a higher concentration of zinc would play a profound role in the inhibition of RdRp. Moreover, pyrithione, zinc ionophore, has been shown to inhibit the replication cycle of SARS-CoV in vitro even in the presence of low level of zinc (2–8 μM zinc) [111], which suggests that zinc can penetrate the viral membrane and suppress viral replication cycle by targeting viral RNA polymerase. The treatment with EDTA, a non-specific zinc chelator, can reverse these effects [111], suggesting a specificity of zinc on RNA polymerase-dependent replication cycle of viruses. One recent report of crystal structural analysis shows that RdRp has zinc binding site in SARS-CoV2 virus [112]. These results suggest that RdRp is a zinc finger protein. The binding of zinc to RdRp may form a highly folded structure of this enzyme, and most likely, results in the decreased binding of the catalytic domain to its substrates, contributing to the inhibition of viral activity. More studies are required to explore these details. Therefore, the implication of zinc would provide a clinical benefit for the treatment of COVID-19 by the inhibition of viral RNA polymerase template binding. However, more studies are required to examine the inhibitory effect of zinc on RdRp in SARS-CoV2 infection.

### 8.2. Viral 3 Chymotrypsin-like (3CL) Protease

Viral 3CL protease, an analogous to 3C cysteine protease molecule that was identified in certain coronaviruses such as picornavirus, is another non-structural protein with a molecular weight of 33 KDa in novel coronaviruses [113]. This internal protease is the key protease which can cleave viral replicase polyprotein at 11 conserved sites containing canonical amino acid cutting sequences (Leu-Gln↓Ser-Ala-Gly) [114,115], leading to the maturation of this protein. Such processing is a very important step for the completion of viral replication cycle and amplification of the infection in SARS-CoV [116,117]. 

Several 3CL protease-targeted inhibitors such as lopinavir and ritonavir have been applied for the intervention of HIV-1 infection. These therapeutic drugs display a positive effect in HIV-1 patients. It seems that this type of viral protein is a potential therapeutic target for the interventional approach of novel coronavirus infections. Moreover, it has been noted that the co-administration of lopinavir/ritonavir, ribavirin, and IFN-α have been utilized for the interventional therapy of MERS-CoV infection with a better clinical outcome [118]. Currently, it has been suggested that lopinavir and ritonavir can be used for COVID-19 treatment. Several clinical trial studies by using lopinavir have shown a clinical improvement in COVID-19 patients [119,120]. Therefore, targeting 3CL protease could be a therapeutic strategy for COVID-19 treatment. 

One early experimental study demonstrated that zinc and zinc-conjugated compound (1-hydroxy-pyridine-2-thione zinc) had a significantly inhibitory effect on 3CL protease activity in SARS-CoV virus by a non-competitive pattern, where the inhibition of zinc is non-competitive to 3CL protease substrate (Ki = 1.1 μM) [117]. The crystallographic study showed 3CL protease had a zinc binding site at the H41-C145 catalytic domain [117]. Recently, one 3D structural analysis study further confirms that zinc has a binding site in viral 3CL protease protein [112], suggesting that 3CL may be a zinc finger protein. Similar to RdRp, the formation of zinc-induced highly folded structure of 3CL may block the access of its catalytic domain to substrates, as discussed above (Section 8.1). Therefore, zinc would have potential as an anti-viral agent by targeting 3CL protease, leading to the inhibition of viral amplification in SARS-CoV and SARS-CoV2 viruses. 

### 8.3. Interferon (IFN)

Human IFN is one group of small, soluble cytokines that are expressed and secreted by the host primarily responsive to viral invasion. Typically, once they are invaded, the virus-invaded cells express and release IFN cytokines in the body. IFN is capable of activating some types of immune cells including monocytes/macrophages, NK cells, and T lymphocytes. These activated immune cells create their anti-viral defense mechanism in a positive feedback loop pattern [121,122]. It is known that IFNs have the activity to inhibit viral replication, as it is named by the protective action of host cells against virus invasions [121,122]. Therefore, IFN play a key role in anti-viral immunity called “anti-viral cytokine” at the early infection stage in the body. Moreover, IFN has also other important activities in the activation of several immune cells such as NK cells, T cells, and monocytes/macrophages as well as the up-regulation of MHC molecules in antigen-presenting cells (APC) to enhance host immune response [121,122]. 

IFN has a wide range of biochemical activities in addition to its known characteristics of anti-viral activity. It is known that IFN participates in other biological processes by the regulation of cell proliferation and differentiation, cell cycle, and host immune response in a positive feedback induction mechanism [121,122,123]. Currently approximately 20 different IFNs are found in humans. Based on its receptor-binding and signal activations, human IFN cytokines are typically defined into three types, namely type I, II, and III. Type I IFNs, (also known as viral IFNs) which include α, β, ε, ζ, δ, and ω, are found to be secreted by all immune cells and virus-invaded cells in response to viral invasion, and bind to a cell surface receptor complex called IFN-α/β (IFNAR/IFNBR) receptors [124,125]. 

A delayed response of the type I IFNs is considered to be highly linked with microbial infections, which include: pneumonia, ARDS, septic shock, and multiple organ failures, eventually causing death [1]. Type II IFNs, which include IFN-γ and are defined as immune IFN are secreted by the majority of immune cells including NK cells, CD4+ T helper cells, and CD8+ T cells, and are able to bind to the IFN-γ receptor (IFNGR) responsive to antigenic or mitogenic stimulations including viral RNA/DNA components. The production of IFN-γ is up-regulated by IL-12/IL-18, and is down-regulated by IL-4/IL-14. IL-12/IL-18 and IL-4/IL-14 are released in macrophages and T helper II cells, respectively [121]. Type III IFNs, such as IFN-λ1, λ2, and λ3, are excreted by epithelial cells and some immune cells and bind to a cell receptor complex consisting of IL10R2 and IFNLR1 [121,126]. Generally, all types of IFNs have anti-viral activity with a similar immune function in the body. 

The anti-viral function of IFNs in the host body has been widely investigated and is highly linked to the inhibition of viral replication life cycle in host cells [121,122,124,126]. Its detailed mechanism is still not fully understood. However, the anti-viral activity of IFNs has been found to be linked with IFN-mediated signaling transduction and transcriptional activation of several genes encoding RNA-dependent protein kinase R (PKR), oligoadenylate synthetase (OAS), RNA-specific adenosine deaminase (ADAR1), protein Mx GTPase, and ribonuclease L (RNase L) through IFN-mediated JAK-STAT/IRF signaling transduction cascade pathways [122]. All these proteins contribute to the inhibition of the viral replication cycle in host cells. 

IFNs are also associated with up-regulation of p21 (a known apoptotic regulator), inducible NO synthase (iNOS), major histocompatibility complex (MHC) proteins, granzyme, and cytotoxic T lymphocyte epitope presentation [122,127,128], all of which are involved in biological processes of anti-viral infection [122,129]. Moreover, IFN also induces FasL and TNF-related apoptosis, leading to the death of virus-invaded cells [130]. For example, one animal study demonstrated that coronavirus infection was associated with increased levels of IFN-γ [127]. This increase was associated with viral clearance along with the up-regulation of MHC molecules [127]. IFN-γ deficiency caused animal death after coronavirus infection [127]. These findings strongly suggest a critical role of IFN-γ against virus infection. 

Aberrant regulations of IFN, especially type II are displayed in COVID-19. Several studies show elevated levels of IFN-γ in COVID-19 [131]. However, other studies report a reduction of IFN-γ levels in patients [132]. These inconsistent findings may be due to the distinct stages and severity of COVID-19 [129]. One report reveals that lower levels of IFN-γ is a risk factor for lung fibrosis in COVID-19. Lung fibrosis is one of the most common causes of lung damage because it causes respiratory failure in COVID-19 [129]. These findings suggest that the dysregulation of IFN-γ may be linked with the pathogenesis of COVID-19. 

IFNs have been considered as antiviral agents and commonly used for the therapeutic treatments of several viral infections [122]. For example, INF-α, as a broad spectrum anti-viral drug, has been routinely used for the treatments of HBV, HCV, and HIV-1 [19,21]. IFN-γ has been used for the treatments of HBV and HCV infections [21,128]. It has been noted that IFNs are used for the treatment of MERS-CoV infection [1]. Therefore, IFNs are suggested for the treatment of COVID-19. Furthermore, the combination of type I IFN-α and type II IFN-γ has been also suggested for the intervention of COVID-19, which has shown a positive effect in SARS-CoV and MERS-CoV infections [21,133]. As a known immune mediator, zinc may regulate the expression of IFNs in the body. It has been noted that zinc deficiency reduces the production of IFN-γ and IL-2. Zinc supplementation reverses these adverse effects [57,134]. 

Our early experimental study demonstrated that zinc deficiency (1 μM zinc) decreased IFN-γ production. Zinc sufficiency (15 μM-50 μM zinc) increased IFN-γ in HUT-78 cells (human malignant leukemia cells) by up-regulating gene expression of T-bet [65], a master transcription factor for the expression of T helper 1-producing cytokines such as IFN-γ, IL-2, and TNF-α/β. This evidence also shows that zinc stimulates the antiviral activity of IFN-α [135]. Therefore, zinc may up-regulate the expression of IFN-γ, leading to the inhibition of viral replication cycle in COVID-19 patients.

### 8.4. Interleukin-6 (IL-6)

IL-6 is another soluble, small protein mediator or cytokine, which belongs to the IL-6 family member. Several different names of IL-6 were used such as B-cell stimulatory factor 2 (BSF2), CTL differentiation factor (CDF), hybridoma growth factor (HGF), and hepatocyte stimulatory factor (HSF), which was based on its function at the time when it was discovered [136,137,138]. IL-6 is expressed and released primarily by many different cells such as monocytes/macrophages, in the response to infections and tissue injury, via the binding of Toll-like receptor (TLR) family proteins to specific microbial molecules termed as pathogen-associated molecular patterns (PAMPs). 

IL-6 is mainly mediated by the activation of NF-κB mediated signaling transduction pathway [137,138]. NF-κB is a master transcription factor responsible for the regulation of the gene expression of a wide variety of pro-inflammatory cytokines, chemokines, and molecules including IL-1, IL-2, IL-2R, IL-6, TNF-α, iNOS, and MCP-1 by a variety of stimuli such as viruses, bacteria, yeasts, LPS, and UV light. Many other cells such as hepatocytes, endothelial cells, muscle cells, and adipose cells also excrete IL-6. 

IL-6 displays pleiotropic effects with pro-inflammatory and ant-inflammatory activities responsive to tissue damage and infection by a context-dependent pattern [137,138,139]. It has been known that IL-6 plays a central role in both innate and adaptive immune response. In addition, IL-6 is involved in other biological processes such as metabolism, regeneration, hematopoiesis, and neural homeostasis [137,138]. IL-6-mediated signaling transduction cascade is activated by its binding to a receptor complex consisting of IL-6 receptor (IL-6R) and glycoprotein 130 (gp130, also called as CD130) [140,141]. 

It has been widely accepted that IL-6 plays a central role in response to inflammation, infection, or tissue injury by the up-regulation of acute phase response (APR) proteins including CRP. APR is a systemic reaction characterized by leukocytosis, fever, elevated vascular permeability, alteration in blood metal and steroid levels accompanied with elevated levels of CRP, which leads to pathogenesis of acute microbial infection [142].

IL-6 is an important cytokine during infection in the body, accompanied with IL-1 and TNF-α, and can be used as an important inflammatory marker for both acute and chronic inflammatory responses [137,138,139,142]. Many clinical observational studies have provided solid evidence to support that IL-6 could be used as a key inflammatory marker for the severity of COVID-19 infection with poor prognosis [143,144,145]. For example, increased levels of IL-6 were found to be highly associated with disease severity of COVID-19 patients, especially with ARDS, suggesting the important role of IL-6 in the pathogenesis of COVID-19. The role of IL-6 in the regulation of viral life cycle is not clearly elucidated. 

Several early studies show that IL-6 may have an anti-viral effect, where exogenous IL-6 was used for the treatment of HBV patients [142]. However, more evidence shows that the hyper-production of IL-6 during viral infection might enhance viral survival by the up-regulation of Th2 cell switch Th0 cells and IL-4 expression. Th2 cell-producing IL-4 cytokine is known to inhibit Th1 cell and IFN-γ excretion, which leads to the downregulation of CD8+ T cells and NK cell for viral clearance in the body. This data suggests that the selective inhibition of IL-6 may provide an additional benefit by the promotion of viral clearance in the body [142]. IL-6 targeted drugs such as anti-IL-6R monoclonal antibody tocilizumab and sarilumab have been approved to alleviate inflammation for the treatment of rheumatoid arthritis [38]. 

Currently, anti-IL-6R drugs have been widely recommended for COVID-19 treatment in clinical practice [47,146,147]. It has been noted that tocilizumab treatment results in a rapid and sustained response as well as clinical improvement in COVID-19 patients with ARDS [39]. Zinc has been shown to inhibit the production of IL-6. For example, our early human subject study demonstrated that zinc supplement (45 mg zinc daily for 6–12 months) decreased plasma level of IL-6, other inflammatory cytokines, and oxidative stress markers in elderly subjects and SCD patients along with the reduction of infection incidence [69,71,78]. Therefore, the application of zinc would provide a positive effect in part by its inhibition of IL-6 production in COVID-19 infection. 

### 8.5. Intracellular Adhesion Molecule (ICAM)

It is known that ICAM, also called as CD54, is a cell surface glycoprotein, which belongs to the immunoglobulin (Ig) superfamily such as antibodies and T cell receptors [148,149]. This molecule is primarily excreted by endothelial cells and some types of immune cells, for example, macrophages and T lymphocytes [149]. Similar to a glycoprotein, ICAM consists of five Ig-like domains [148,150]. The domains 1 and 2 have been identified as the binding site with microbial pathogens including viruses. Other separate domains of ICAM can bind its cognate ligand integrin (LEF1, or known as CD18/CD11a) on leukocytes, which triggers the migration of immune cells to the infective site [148,149,150]. ICAM is an immune modulator that viruses bind to and affect host cells like alveolar cells in respiratory system. ICAM is primarily involved in the immune response by modulating the recruitment of immune cells toward the site of infection. 

Currently, ICAM has been widely considered as a primary entry receptor for several viruses such as HIV-1, Coxsackievirus A21, Influenza A, and human rhinovirus [150,151,152]. Rhinovirus is one of the most important etiological pathogens of upper respiratory tract infections (usually called as the common cold). Coxsackievirus A21 and rhinovirus utilize ICAM acting as a viral receptor for host cell adhesion, which leads to viral invasion. The up-regulation of ICAM expression can be induced by these viruses via the activation of NF-κB-mediated signaling pathway in respiratory epithelial cells [150,152]. Moreover, elevated ICAM level is observed in patients with ARDS [153,154].

Interestingly, it is observed that COVID-19 patients show an increase in serum ICAM level. One recent study reports that the concentration of serum ICAM is highly linked with COVID-19 severity [155]. Importantly, the prognosis from COVID-19 patients with severe illness is linked with the lower concentrations of several inflammatory cytokines including ICAM, TNF-α, and CRP [155], suggesting that ICAM might display an important role in the pathogenesis of COVID-19. The selective inhibition of ICAM might provide a therapeutic strategy for the treatment of COVID-19. Currently, ICAM inhibitors such as A-205804, RWJ50271, and BMS-688521 are suggested to have potential for COVID-19 treatment [156]. These clinical trials of ICAM inhibitors for COVID-19 are still under investigation. 

Although the detailed mechanism of ICAM in the pathogenesis of COVID-19 is not clear, it is reasonable to speculate that ICAM may act as an entry co-receptor for SARS-CoV2 infection, which is similar to the role of ICAM in the infections of rhinovirus and other viruses. However, more research is required to study the detailed mechanism of ICAM in COVID-19. Our early clinic study demonstrated that zinc treatment significantly reduced the severity of the common cold along with the inhibition of sICAM production in human subjects [87], which suggests that zinc may have a potential as anti-viral agent, in part by the inhibition of ICAM.

### 8.6. Natural Killer (NK) Cells

NK cells are one class of lymphoid cells that exhibit an absence of TCR on their cell surface. They participate in the innate immune response. NK cells are originated from hematopoietic stem cells (HSC) in the bone marrow and mature in the thymus gland. This group of lymphoid cells plays an essential role in immunity against microbial infections and tumor cells [130,157,158]. Aptly named, NK cells can kill virus-invaded cells and tumor cells through MHC I like molecule tuning or education [130,157,158]. 

NK cells also excrete several important cytokines including INF-γ and IL-2 that promote anti-viral adaptive immune response. Its cytolytic and cytotoxic activity can be modulated by the recognition of virus-derived products or viral peptide-mounted MHC I or MHC I like molecules. It is known that NK cells eradicate viruses or virus-infected cells by three main mechanisms, (1) the excretion of IFN-γ, an anti-viral cytokine as described above; (2) the secretion of cytolytic granules containing granzyme and perforin; (3) through the use of death-receptor-mediated apoptosis [130,157,158]. 

It is known that aberrant modulation of NK cell functions, for example, lower counts and activities of NK cells, in part impacted by an activated cytokine storm during virus infection is highly linked to the severity of COVID-19 [130,157,158]. For example, it has been discovered that the low level of NK cells, including serum cytotoxic effector molecules like perforin and granzyme A, is linked with the severity of COVID-19 [159,160,161,162]. The enhancement of NK cell function would provide a positive effect on viral clearance in the body. 

A large number of studies have clearly revealed that zinc positively modulates NK cells [57,163,164]. Zinc deficiency reduces NK cell activity. Zinc supplementation reverses this adverse effect [57,59,163,165,166]. Early studies demonstrated that the cytolytic activities of NK cells were suppressed during zinc depletion. This suppression was linked with a decrease in the recognition of MHC-1 molecule, and NK cell activity was higher when zinc was supplemented to cells [167,168,169]. Zinc supplementation up-regulated cytolytic activity of NK cells and CD8+ T cells toward the infection site [167,168]. One early human study revealed that zinc supplementation up-regulated NK cell activity and IL-2 expression along with the reduction of infection rate in elderly subjects [170]. One recent experimental study shows that the up-regulation of NK cell function induced by zinc is linked with the enhancement of perforin expression [171]. 

IL-2 is known to enhance NK cell proliferation and activation. Interestingly, the evidence from experimental and human subject studies distinctly indicates that zinc can increase IL-2 production in lymphocytes [66,67]. Therefore, the up-regulation of IL-2 by zinc could contribute to the enhancement of NK cell function for viral killing capacity. These data suggest that zinc can enhance NK cell activity by the regulation of IL-2 and perforin as well as the MHC I recognition, which contribute to viral clearance in the body. 

### 8.7. CD4+ and CD8+ T Cells

CD4+ and CD8+ T cells, two main subtypes of mature T lymphocytes that express of T cell receptor (TCR) on their cell surface, belong to one of three classes of lymphoid cells that are originated from HSC in the bone marrow and mature in the thymus gland [172,173,174]. Physiologically, CD4+ T cells and CD8+ T cells play a pivotal role in cellular immunity by protecting host cells from microbial infections and tumors [172,173,174].

CD4+ T cells (also known as T helper cells, or CD4+ T helper cells) can be identified into several distinct subgroups such as T helper type I (Th1), Th2, Th17, T regulatory cells (Treg) cells, T follicular helper cells, and memory T cells, which are based on their functions and protein profiles. Th1 and Th2 cells are two major subgroups of CD4+ T cells. Th1 cells express and release IFN-γ regulated by T-bet transcription factor responsive to a variety of stimuli including viruses [172,173,174]. Th1 cells also can excrete IL-2, which has been defined as a T cell growth factor by the regulation of T cell proliferation and differentiation. Th2 cells can predominantly produce IL-4, IL-6, IL-10, and other cytokines regulated by GATA3, a key transcription factor [175,176,177]. It is known that Th2 cells play an important role in maintaining maturation of B cells into antibody-producing plasma cells and memory B cells as well as in the activation of cytotoxic T lymphocytes [175,176,177]. 

Cytotoxic CD8+ T cells, another major subtype of T lymphocytes, are also known as cytotoxic T lymphocytes (CTL), T killer cells, or killer T cells, which are cytotoxic and can directly eradicate virus-invaded cells and tumor cells through the recognition of MHC I molecules by the secretion of granzyme- and perforin-containing cytolytic granules [172,173,174]. CD8+ T cells can also excrete several cytokines including IFN-γ and IL-2 to enhance immune defense response [172,173,174]. 

Early retrospective studies indicated that SRAR and MERS patients had a significant reduction of T cells counts including CD4+ and CD8+ cells during the acute infection phase [178]. It has been noted that COVID-19 patients present the dysregulation of cellular immune response following the pro-inflammatory phase, characterized by sustained and substantial decrease in peripheral lymphocyte counts, specifically CD4+ and CD8+ T cells [37,38,132,159,160,161,162]. The degree of malfunction of T lymphocytes has been found to be linked with the severity of COVID-19 patients [131,159,160,161,162]. 

It has been noted that maintaining and boosting normal functions of the cellular immune response, for example, CD4+ and CD8+ T cell activities, are critical in anti-viral immunity in the body [122]. One early study demonstrated that zinc supplementation (10 mg daily and 20 mg during diarrhea) reduced the incidence of acute lower respiratory infection and increased the count of CD4+ cells in preschool children [73]. Our human studies indicated that zinc deficiency decreased the relative level of CD8+/CD73+ T lymphocytes, a precursor of cytolytic T cells. Zinc supplementation can reverse this adverse effect [61,62]. Other human studies have demonstrated that zinc supplementation increases the numbers of CD4+ and CD8+ T cells along with the improvement of cell-mediated immune response in elderly subjects [57,179,180]. One clinical study showed that zinc treatment reduced the infection incidence and increased CD4+ T cell counts and thymulin in HIV patients [181]. These findings suggest that zinc might have an anti-viral effect in part by the regulation of CD4+ and CD8+ T cells.

## 9. Adverse Effect of Zinc Treatment/Supplementation

Generally, zinc is nontoxic, which is similar to other trace mineral nutrients in the body. The main source of zinc intake by humans is from foods such as meats, fish, eggs, seeds, seafood, certain vegetables, whole cereal, and beans [182,183]. The recommended daily allowance (RDA) of zinc is 11 mg for men, and 8 mg for women in the US [184,185]. The median lethal dose (LD50) is estimated to be 27 g for humans based on animal studies [68], which is remarkably lower than the LD50 value of zinc. Between 15–50 mg of zinc supplementation for over-extended periods of time has been used in different populations including elderly subjects and SCD patients without any adverse effect observed [69,71,78,186]. 

It has been identified that high dose of zinc intake may cause adverse gastrointestinal effects such as abdominal pain, nausea, and vomiting, which is often tolerated. For example, in our early human subject studies, we did not observe any adverse effects in elderly subjects and SCD patients who followed 45 mg of oral zinc intake daily for 6–12 months [71,78]. In our zinc clinical trials, we also did not observe any adverse effects in patients with the common cold who followed high dose of zinc treatment (13 mg zinc, 7 times daily, equivalent to 91 mg zinc daily) for more than 11 days, compared to the placebo-control patients [86,87], which is consistent with the findings from other reports [89,90]. However, one clinical trial shows minor side effects of nausea and bad-taste reaction following higher dose of zinc treatment (13.3 mg zinc every 2 h while awake daily) for therapy of the common cold. Furthermore, high dose of zinc-induced copper deficiency syndrome has received a great amount of attention [68,187,188]. 

Several animal reports indicate that high dose of zinc intake over an extended period of time is frequently linked with high risk of copper deficiency in rats and mice [189,190], which is induced by a competitive absorption between zinc and copper in the intestinal mucosa [189,190,191,192]. Several human case reports confirmed these findings [193,194,195]. For example, one SCD patient took therapeutic dose of zinc (150 mg daily) during hospitalization followed by 110 mg of zinc treatment for more than one year, which later led to the development of copper deficiency syndrome like lower levels of serum copper and anemia. This zinc-induced copper deficiency was easily corrected by the treatment of 1 mg copper daily for 2 weeks [195]. Similar findings were reported in another case of the Hallervorden-Spatz syndrome patient who took a therapeutic dose of zinc (100 mg) daily for more than five years [194]. Conversely, one human subject study demonstrated that 26 of 47 healthy volunteers had symptoms of abdominal cramp, nausea, and vomiting following the supplementation of 150 mg zinc daily for six weeks [196]. Plasma copper level was not significantly altered [196]. After 12 weeks of zinc supplementation, only female volunteers had a 14% reduction of serum ceruloplasmin (from 13.0 to 11.3 U/mL) [197]. These adverse effects were related to lower body weight [197]. These data suggest that zinc is safe when 50 mg or less daily is used for long term supplementation while 50–150 mg daily for a short term is safe for therapeutic purpose. 

In one early AREDS study supported by the National Institute of Health/the National Eye Institute in the USA, which examined the effect of antioxidants and zinc on age-related macular degeneration (AMD) in the elderly subjects, 2 mg of copper daily was added to the zinc supplemented group (80 mg zinc daily) in order to prevent copper deficiency development. This study was performed for more than 6 years without any adverse effects observed in the zinc supplemented group [198]. These findings suggest that copper deficiency induced by high dose of zinc administration can be prevented by the co-supplementation of 2 mg copper with zinc. The copper status in the body may be required for monitoring based on the dose of zinc and the period of time when it is given. Generally, zinc is nontoxic, and its common adverse effects usually include gastric discomfort, nausea, and vomiting. The detrimental effect of copper deficiency induced by a large amount of zinc supplementation/treatment would be preventable by the co-administration of 2 mg of copper. 

## 10. Conclusive Remark and Future Prospective

In summary, the COVID-19 pandemic was caused by a novel coronavirus SARS-CoV2, which is still ongoing with a high morbidity and mortality rate. The pathogenesis of COVID-19 is highly linked with suppressed cellular immune function, aberrant immune response, and hyper-inflammation, which leads to an activated cytokine storm and results in ARDS. The higher levels of key inflammatory cytokines/chemokines including IL-1, IL-6, TNF-α, MCP-1, and CRP are highly linked with severity of the disease. Dysregulations of T cells including CD4+ and CD8+ T cells are reported in COVID-19 and are linked with the high risk of severity of this disease. 

To date, there is no proven and effective therapeutic treatment for COVID-19, which provides a significant challenge in exploring a better therapeutic strategy for the treatment of this disease. It is known that zinc is an essential trace mineral element in human health by the involvement of a wide variety of biological processes like the innate and adaptive immune responses. The application of zinc as an anti-inflammatory and anti-oxidant agent has been on the rise for use in human health and disease. Early clinical data showed that the application of zinc can decrease the severity of viral infections including rhinovirus infection (the common cold) in human adults and the incidence of infection in children and elderly subjects through down-regulation of inflammatory cytokines/chemokines and oxidative stress markers. 

New evidence reveals that the triple combination of HQC, azithromycin, and zinc reduces rates of hospital admission and death of COVID-19. These findings suggest that zinc may have anti-viral activity that could be applied for treatment of viral infections including COVID-19 but its detailed mechanism is not fully understood. Despite this, the results from increasing numbers of studies suggest that zinc may have an inhibitory effect on some viral infections including SARS-CoV2, potentially through the regulation of viral RdRp and 3CL protease proteins as well as host IFNs, IL-6, ICAM cytokines, NK cells, and CD4+/CD8+ T cells as shown in Figure 1. Currently, it has been reported that more than 18 clinical trials using zinc with a wide range of doses and treatment periods are registered in current clinical practice. Therefore, the application of zinc as a potent anti-viral agent along with its anti-inflammatory and anti-oxidant properties may provide meaningful clinical benefit for the treatment of COVID-19. Well-designed clinical trials are required to further understand the clinical benefit of zinc as an anti-viral agent for the treatment and/or prevention of COVID-19 infection in the future.

## Figures and Tables

**Figure 1 antioxidants-11-01862-f001:**
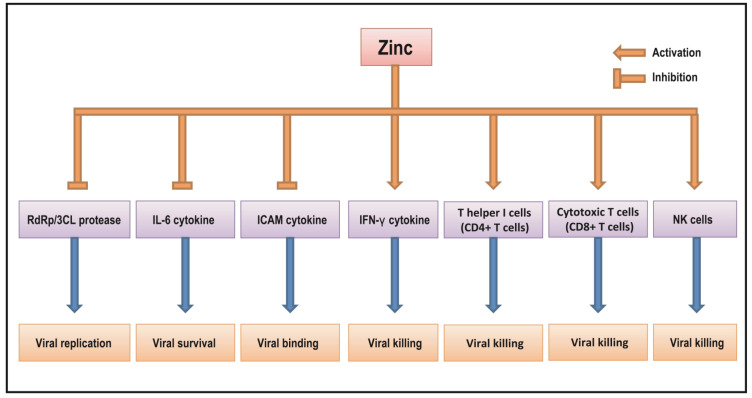
The putative mechanism of zinc action as an anti-viral agent in COVID-19.

**Table 1 antioxidants-11-01862-t001:** Characteristics of clinical manifestations in COVID-19 patients.

Severity	Symptoms and Signs	Patients %
**Mild**	Low-grade fever, dry cough, fatigue, nausea, vomiting, new loss of taste and smell.	≤80%
**Moderate**	High-grade fever, deep dry, fatigue, muscle or body aches.	15–20%
**Severe**	High-grade fever, deep dry, fatigue, body aches, shortness of breath, chest pain, confusion, and purple lips.	≤5%
**Critical**	ARDS, multiple organ failures such as lung, heart, and kidney failures, leading to death of patients.	≤1–2%

**Table 2 antioxidants-11-01862-t002:** Clinical outcomes or benefits of zinc treatment in COVID-19.

Authors	Study Types	Patients and Experiment Design	Clinical Outcomes or Benefits
Abd-Elsalam et al., 2021 [96]	A randomized study	191 hospital patients; combination of HCQ with zinc (*n* = 96; mean age 43.48 + 14.62; 50 mg zinc, twice daily, 15 days) vs. HCQ only (*n* = 95; mean age 43.64 ± 13.17)	No benefits were found on ventilation usage and death rate. Limitation: potentially high phytate diet interrupting zinc absorption in the Middle East area.
Carlucci et al., 2020 [97]	A retrospective/Cohort study	932 patients; combination of HCQ and azithromycin plus zinc (50 mg zinc, twice daily, 5 days) (*n* = 411; mean age 63.19 + 15.18) vs. combination of HCQ and azithromycin (*n* = 521; mean age 61.83 + 15.97)	Bivariate regression analysis showed that the addition of zinc increased the frequency rate of discharging home; decreased the rates of mortality, ventilation usage and ICU admission.
Derwand et al., 2020 [92,93]	A retrospective/Cohort study	141 out-patients; median age 58; combination of HCQ and azithromycin plus zinc (50 mg zinc, once daily, 5 days)	Combination with zinc decreased the rates of hospital admission (2.8% vs. 15.4%) and mortality (0.7% vs. 3.4%), compared to independent public reference data of 377 COVID-19 patients in the same community setting (used as untreated control).
Elalfy et al., 2021 [98]	A non-randomized controlled study	113 out-patients with mild symptoms; mean age 51; combination of nitazoxanide, ribavirin, and ivermectin plus zinc (*n* = 62; 30 mg zinc; twice daily; 15 days) vs. supportive treatment (*n* = 51)	Combination with zinc decreased the rates of COVID-19 virus clearance and symptoms.
Finzi and Harrington, 2021 [99]	A retrospective review study	28 out-patients; 15–23 mg zinc daily; minimum of 10 days	Zinc improved symptom scores. None of patients were admitted to hospital after treatment. Of the 28 patients, 26 were asymptomatic; 2 patients were still fatigued after zinc treatment. Limitation: absence of blinding and control group.
Frontera et al., 2020 [100]	A retrospective/Cohort study	3473 in-patients, media age 64; combination of HCQ and zinc (*n* = 1006; 50 mg zinc, once or twice daily; minimum of 4 days) vs. no treatment of HCQ and zinc (*n* = 2467)	Combination with zinc significantly decreased in-hospital mortality rate by 24% and increased the frequency rate of discharging home by 7.4%.
Patel et al., 2021 [101]	A randomized controlled study	33 hospital patients with zinc deficiency; zinc treatment (*n* = 15; mean age 59.8 + 16.8; 0.24 mg zinc/kg/day; iv injection; maximum of 7 days) vs. placebo patients (*n* = 18; mean age 63.8 + 16.9).	No clinical benefit on blood oxygenation and mortality rate was found. However, zinc injection corrected zinc deficiency in patients. Limitation: small sample size.
Thomas et al., 2021 [102]	A randomized controlled study (open label)	214 out-patients (mean age 45.2 + 14.6); 4 groups (1:1:1:1); (1): 50 mg zinc, daily; 10 days; (2): Vitamin C, 8000 mg; daily; 10 days; (3): Combination of zinc and vitamin C; (4): Neither zinc nor vitamin C.	No benefit was found on the duration of symptoms. The study was terminated due to a low conditional power.

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
