# Peer review of "The Mechanisms of Zinc Action as a Potent Anti-Viral Agent: The Clinical Therapeutic Implication in COVID-19"

_antioxidants, 2022, doi:10.3390/antiox11101862_

Round 1
Reviewer 1 Report
The authors present a timely review about zincs role in viral infection and possible implications with SARS CoV2. There are a few concerns that need to be addressed before this review can be published.
- The abstract is very long and should be reduced by atleast 50%; the abstract is supposed to be a summary.
- Throughout the review, there are many formatting and typos - the authors need to thoroughly proof read this manuscript.
- Line 65 re-word 'This virus...' This paragraph (line 64-79) should be revised to improve the English.
- Line 113-117 also needs improving as the information is listed rather than written with a smooth flow. Reference missing Line 114.
- Line 184 - it would be good to mention zinc ejectors for HIV here https://pubmed.ncbi.nlm.nih.gov/8863808/, https://pubs.acs.org/doi/10.1021/jm9802517, https://pubmed.ncbi.nlm.nih.gov/29107765/, https://pubmed.ncbi.nlm.nih.gov/29326078/ (among many others), and cross species potential https://pubmed.ncbi.nlm.nih.gov/25702849/, https://pubmed.ncbi.nlm.nih.gov/31101470/.
- The review while useful needs to tone down the implication that zinc supplements can help against viral infection in the 'meaningful' way. They can assist, but this is very different to actually being part of a treatment regime. This is evident in the abstract and conclusion section.
Author Response
To Review 1
We thank the reviewer 1 for his/her comments and suggestions so that we can improve the quality of our manuscript.
Comment # 1: The abstract is very long and should be reduced by at least 50%; the abstract is supposed to be a summary.
Response: The abstract has been significantly reduced, approximately 50% in the Abstract section (Please see page 1 in the revised manuscript).
Comment # 2: Throughout the review, there are many formatting and typos - the authors need to thoroughly proof read this manuscript.
Response: We apologize for these mistakes. A native English colleague has thoroughly proof-read this manuscript. We have already made corrections. Please see the highlights by Track Change in the revised manuscript.
Comment # 3: - Line 65 re-word 'This virus...' This paragraph (line 64-79) should be revised to improve the English.
Response: We already revised this paragraph. Please see Line 67-72 and Line 79 in our revised manuscript.
Comment # 4: - Line 113-117 also needs improving as the information is listed rather than written with a smooth flow. Reference missing Line 114.
Response: We have made a table for this information (Please see Table 1). We added missed references in Line 123 in the revised manuscript.
Comment # 5: - Line 184 - it would be good to mention zinc ejectors for HIV here https://pubmed.ncbi.nlm.nih.gov/8863808/, https://pubs.acs.org/doi/10.1021/jm9802517,
https://pubmed.ncbi.nlm.nih.gov/29107765/, https://pubmed.ncbi.nlm.nih.gov/29326078/
(among many others), and cross species potential https://pubmed.ncbi.nlm.nih.gov/25702849/,
https://pubmed.ncbi.nlm.nih.gov/31101470/.
Response: We have discussed the role of zinc ejection for HIV and, potentially for COVID-19 in Line 201-219 in our revised manuscript.
Reviewer 2 Report
Zinc homeostasis in COVID-19 is discussed in this article. The use of zinc to guard against COVID-19 infection was mentioned. Zinc acts as a powerful antiviral drug by targeting viral RNA dependent RNA polymerase and three chymotrypsin-like proteases, as well as host IFN, IL-6, ICAM cytokines, NK cells, and CD4+/CD8+ T cells. Finally, the potential side effects of zinc therapy in humans were investigated.
Add a table showing the clinical trials that used zinc in treatment of COVID-19, the enrollment, type of trial, epidemiological data and the outcomes.
Discuss in more details the essential rules of zinc in the structure stabilization and molecular dynamics of viral enzymes
Discuss the impact of zinc deficiency on the progress of respiratory infections in a separate section
What is the rule of zinc in prophylaxis of COVID-19?
Explain the rule of zinc in section 2 “2. SARS-CoV2 virus and its pathogenesis ”
most sections needs to be supported by expressive figures.
Author Response
To Reviewer 2
We thank the reviewer 2 for his/her comments and suggestions so that we can improve the quality of our manuscript.
Comment # 1: Add a table showing the clinical trials that used zinc in treatment of COVID-19, the enrollment, type of trial, epidemiological data and the outcomes.
Response: We have added one table showing the clinical outcomes/benefits of zinc treatment and combination with zinc from recent clinical trials in COVID-19 patients (Please see Table 2).
Comment # 2: Discuss in more details the essential rules of zinc in the structure stabilization and molecular dynamics of viral enzymes
Response: We have discussed further details of the role of zinc in the structure stabilization and molecular dynamics of viral enzymes in Line 396-399 and Line 427-430 in our revised manuscript.
Comment # 3: Discuss the impact of zinc deficiency on the progress of respiratory infections in a separate section. What is the rule of zinc in prophylaxis of COVID-19?
Response: We have discussed the impact of zinc deficiency on the progress of respiratory infections in a separate section, namely the Section 5, Line 273-283.
Comment # 4: What is the rule of zinc in prophylaxis of COVID-19?
Response: We have discussed the rule of zinc in prophylaxis of COVID-19 in Line 346-361.
Comment # 5: Explain the rule of zinc in section 2 “2. SARS-CoV2 virus and its pathogenesis ” most sections needs to be supported by expressive figures.
Response: We have explained the role of zinc in section 2 “2. SARS-CoV2 virus and its pathogenesis ”, Line 105-110. We also add 2 tables to express information rather than written (Please see Tables 1 and 2).
Round 2
Reviewer 1 Report
The authors have addressed the concerns, a final proof read would be useful.
Reviewer 2 Report
Accept